# Separation of Neodymium (III) and Lanthanum (III) via a Flat Sheet-Supported Liquid Membrane with Different Extractant-Acid Systems

**DOI:** 10.3390/membranes12121197

**Published:** 2022-11-28

**Authors:** Lin Li, Ben Yu, Krystal Davis, Aaron King, Mauro Dal-Cin, Andrzej Nicalek, Naiying Du

**Affiliations:** National Research Council of Canada, Ottawa, ON K1A 0R6, Canada

**Keywords:** supported liquid membrane, solvent extraction, Neodymium(III), Lanthanum(III), extractant

## Abstract

The increasing demand for neodymium (Nd) permanent magnets in electric motors has revived research interest of Nd recovery and separation from other rare earth elements (REEs). Typically, Nd/La separation is necessary for Nd recovery from primary ores and secondary resource recycling. This research used a flat sheet-supported liquid membrane (FSSLM) with different extractant-acid systems to extract Nd from a Nd/La mixture. The recovery and separation of Nd/La with 204P-H_2_SO_4_, 507P-HCl, and TBP-HNO_3_ were discussed. The results showed effective Nd recovery and promising Nd/La selectivity could be achieved in the 507P-HCl system, compared to 204P-H_2_SO_4_ and TBP-HNO_3_. The addition of citric acid to the feed solution was effective for pH buffering but did not improve the Nd transport or Nd/La selectivity. Long-term stability of the 507P-HCl extractant system was demonstrated by extending the processing time from 6 h to 6 days.

## 1. Introduction

Due to their superior physical properties, rare earth elements (REEs) have critical applications in modern technologies, such as wind turbines, LED light bulbs, permanent magnets, catalysts, and other products [1,2,3]. Nd_2_Fe_14_B permanent magnets are the most powerful and widely commercialized permanent magnets, with increasing demand in the market. They offer the best value for cost and the strongest performance [4,5,6]. Monazite and bastnäsite minerals are the primary resources of Nd. Other sources include Ni-metal hydride (Ni-MH) batteries [7,8,9]. A typical Nd recovery process involves leaching and purification or REEs separation. For example, for waste Ni-MH batteries leaching in acidic media, Ni(II), Co(II), and Mn(II) are first recovered with multiple chemical methods, while other elements, including La(III) and Nd(III), are left in the leach liquor, which is then subjected to further separation technology [10].

The efficient separation of REEs has been an intractable problem, as REEs present similar chemical and physical properties [11]. Among all the developed separation technologies, the solvent extraction (SX) process is used commercially to separate REEs, producing high-purity single rare earth products. The most common commercial extractants reported are cation exchangers (or acidic extractants), solvation extractants (or neutral extractants), and anion exchangers (or basic extractants) [5,12,13,14,15]. Acidic organophosphorus extractants, such as di-2-ethylhexyl phosphoric acid (204P) [16] and mono-(2-ethylhexyl) phosphonic acid mono-2-ethylhexylester (507P), have been studied the most [12,16,17,18]. Other common extractants reported in the literature are tri-n-butyl phosphate (TBP) [10,19,20] and tri-n-octyl phosphine oxide (TOPO) [10,20]. The conventional SX method to separate rare earth has the advantage of highly efficient recovery and processing of large volumes of REEs solutions. However, it also produces large volumes of waste solutions and the extractant dissolves or evaporates during the process, which causes severe environmental pollution [10].

An alternative technology is the supported liquid membrane (SLM) separation technique, which has shown promise and overcome some major drawbacks of conventional SX. SLM technology requires significantly less organic solvent than SX, while it also has the advantage of reduced energy consumption and operating costs [21,22]. The standard configuration of SLM includes flat sheet-supported liquid membrane (FSSLM) and hollow fiber-supported liquid membrane (HFSLM). HF modules offer high surface area to unit volume ratios, up to 500 m^−1^ [23]. Currently, commercial HF modules are available up to 220 m^2^. Recent advances in supported liquid membrane technology broaden its application, such as separating citric and lactic acid in an aqueous solution [24], increasing the concentration for analytical chemistry [25].

Flat sheet-supported liquid membrane (FSSLM) is chosen in this research for Nd recovery from La/Nd binary solution with commercially available acidic or neutral organophosphorus extractants, including 204P, 507P, and TBP. The Nd/La separation has been systematically studied in our recent research [26]. This study investigated Nd/La recovery and separation in the 204P-H_2_SO_4_ and TBP-HNO_3_ systems. In addition, the effect of the auxiliary agent, citric acid, and longtime stability test on Nd recovery and separation of La/Nd have been demonstrated.

## 2. Material and Methods

The chemical reagents used for the REEs feed solution were AR-grade La_2_O_3_ (≥99.9%, Sigma-Aldrich, Oakville, Canada) and Nd_2_O_3_ (99.9%, Sigma-Aldrich, Oakville, ON, Canada) powders. For preparing feed solutions, the REE oxide was dissolved by concentrated acids (HCl, H_2_SO_4,_ and HNO_3_) at 80 °C, diluted, and the solution pHs adjusted to the desired values. Strip solutions were prepared by diluting the respective acid to the desired concentration using de-ionized water (<20 uS/cm). The support was microporous polypropylene membranes (Celgard^®^ 2500, Charlotte, NC, USA) with an effective surface area of 5 × 5 cm^2^, 55% porosity, and 25 µm thickness. All the extractants, including 507P, 204P, and TBP, were purchased from Kopper Chemical, Chongqing, China. Exxsol^TM^ D80, ExxonMobil, USA was used as the diluent for the extractant.

Batch tests using FSSLM to separate Nd/La were conducted in a custom two-compartment cell, and the setup is presented in Figure 1. For the preparation of the separation tests, the support membrane was first immersed in a predetermined % (*v*/*v*) of 204P (507P; TBP)/D80 mixture followed by sonication for 10 min in an ultrasound bath (Cole-Parmer 3 Liter Ultrasonic Cleaner, Quebec, QC, Canada). The application of ultrasound enhanced the absorption of the extractant into the membrane pores. Before each recovery test, the organic loaded membrane was taken out after excess carrier solution on the surface of the membrane was blotted with lint free tissue. Then, the soaked membrane was installed between the two compartments containing the 250 mL feed and strip solution.

During Nd/La transport, the feed and strip solutions were stirred at 350 rpm for 6 h at room temperature. Feed and strip samples (0.25 mL) were withdrawn at 0, 0.5, 1, 1.5, 2, 3, 4, 5, and 6 h. Nd and La concentrations were measured by ICP-OES (Inductively Coupled Plasma-Optical Emission Spectrometry, Perkin Elmer Optima 3000) to determine their recovery efficiencies. The SLM stability test was six days, with sampling times at 1, 4, 24, 48, 72, 96, 120, and 144 h.

## 3. Results and Discussion

The transport of Nd and La by SLM using acidic organophosphorus extractants, such as 204P and 507P (abbreviated as (HA)_2_), can be seen as a cation exchange process. During the process, hydrogen ions are released from the extractants while the Nd and La occupy the vacancies, as shown in Equations (1) and (2).
(1)$${\mathrm{La}}^{3+}+3{\left(\mathrm{HA}\right)}_{2}↔{\mathrm{LaA}}_{3}{\left(\mathrm{HA}\right)}_{3}+3H^{+}$$
(2)$${\mathrm{Nd}}^{3+}+3{\left(\mathrm{HA}\right)}_{2}↔{\mathrm{NdA}}_{3}{\left(\mathrm{HA}\right)}_{3}+3H^{+}$$

As for the TBP reacting with La or Nd, the equation can be written as:(3)$${\mathrm{La}}^{3+}+3\;\mathrm{TBP}↔{\left[\mathrm{La}{\left(\mathrm{TBP}\right)}_{3}\right]}^{3+}$$
(4)$${\mathrm{Nd}}^{3+}+3\;\mathrm{TBP}↔{\left[\mathrm{Nd}{\left(\mathrm{TBP}\right)}_{3}\right]}^{3+}$$

The recovery fraction was used to quantify the Nd and La recovery efficiency, representing the Nd or La distribution ratio in the strip and feed solution, as shown in Equations (5) and (6):(5)$${\mathrm{Recovery}\%}_{{\mathrm{La}}^{3+}}=\frac{{\mathrm{La}}^{3+}\left(\mathrm{strip}\right)}{{\mathrm{La}}^{3+}\left(\mathrm{initial}\right)}\times 100\%$$
(6)$${\mathrm{Recovery}\%}_{{\mathrm{Nd}}^{3+}}=\frac{{\mathrm{Nd}}^{3+}\left(\mathrm{strip}\right)}{{\mathrm{Nd}}^{3+}\left(\mathrm{initial}\right)}\times 100\%$$
where ${\mathrm{Nd}}^{3+}\left(\mathrm{strip}\right)$ and ${\mathrm{Nd}}^{3+}\left(\mathrm{initial}\right)$ represent the final Nd(III) concentration in the strip solution and the initial concentration in the feed solution, respectively, and likewise for ${\mathrm{La}}^{3+}\left(\mathrm{strip}\right)$ and ${\mathrm{La}}^{3+}\left(\mathrm{initial}\right)$. The Nd(III)/La(III) selectivity is determined as the ratio of the respective recoveries:(7)$$\alpha_{{\mathrm{Nd}}^{3+}/{\mathrm{La}}^{3+}}=\frac{{\%\;\mathrm{Recovery}}_{{\mathrm{Nd}}^{3+}}}{{\%\;\mathrm{Recovery}}_{{\mathrm{La}}^{3+}}}$$

### 3.1. Nd/La Separation in the 204P-H_2_SO_4_ System

Recovery and separation of Nd/La in different extractant-acid systems was investigated, including 204P-H_2_SO_4_ and TBP-HNO_3_ in this work, which are compared to 507P-HCl in our previous research [26]. The 204P-H_2_SO_4_ system has been widely used for REEs recovery in aqueous solution, since the distribution ratios of the REEs are markedly different from other impurities in the leach liquor. It has also been generally applied for rare earth separation [27]. For example, the bastnäsite ore, the primary resource of REEs in China, is typically roasted with sulfuric acid, followed by leaching with water or diluted sulfuric acid solution [28]. Then 204P extractant was used to enrich the REEs. In this work the Nd/La separation in the 204P-H_2_SO_4_ system was studied concerning the effect of pH in the range of 2 to 4. The feed phase was a 250 mL solution with 1000 mg/L of Nd and La, each, at initial pHs of 2, 3, and 4. The supported membrane was loaded with 20% (*v*/*v*) 204P/D80, and the strip solution was 1.5 M H_2_SO_4_. The results of Nd/La recovery and separation in the 204P-H_2_SO_4_ system are shown in Figure 2.

From Figure 2a,b, the Nd recovery increased with increasing pH in the range of 2 to 4. The Nd recovery after 6 h was 58.17, 63.49, and 71.23% from initial feed solution pHs of 2, 3, and 4, respectively. Similarly, the La recovery slightly increased with increasing feed solution pH. The La recovery after 6 h, was 17.50, 18.83, and 20.82% for initial feed solution pHs of 2, 3, and 4, respectively. The recovery rates for both Nd and La are nearly constant, suggesting zero^th^ order kinetics with respect to the concentration of the respective elements, in the time span and recovery studied. The overall recovery rates were 1.59, 1.70, and 1.96 gmoles/m^2^/day at initial pHs of 2, 3, and 4 respectively.

The selectivities calculated based on the Nd and La recovery at 6 h are shown in Figure 2c. As can be observed, the selectivity marginally increased with initial feed solution pH for Nd/La separation in the 204P-H_2_SO_4_ system. Compared to Nd/La separation in the 507P-HCl system [26], the Nd recovery efficiencies are much lower in the 204P-H_2_SO_4_ system conducted under the same experimental conditions. The Nd recovery after 6 h using 20% (*v*/*v*) 507P-HCl with an initial feed solution pH of 3 was 93.10%, and the Nd/La selectivity of 6.39 compared to the 204P-H_2_SO_4_ system’s selectivity of 3.37.

### 3.2. Nd/La Separation in the TBP-HNO_3_ System

TBP has been widely used for metal separation, and is more efficient in nitrate solution than in chloride and sulfate solution [29]. TBP has also been applied for extraction and purification in REE production from their ores. For instance, TBP has been used complexed with nitric acid solution to extract yttrium, lanthanum, cerium, and neodymium from aqueous solution after apatite concentrate is leached [28]. More than 98% of the apatite REEs can be recovered using TBP in 0.4 M HNO_3_ [30].

In this study, Nd/La recovery and separation were examined with the TBP-HNO_3_ system by FSSLM. The effect of pH (pH 2–4) using 20% (*v*/*v*) TBP showed negligible Nd and La recovery, results not shown. The effect of TBP concentration on Nd/La recovery and separation was evaluated with a feed solution of 1000 mg/L La (III) and Nd(III), each, at an initial pH of 3 and 3 M HNO_3_ strip solution. The extractant composition was 20%, 40%, or 60% (*v*/*v*) TBP/D80. The recovery and separation of Nd/La results in 6 h are shown in Figure 3.

When using 20% (*v*/*v*) TBP, Nd and La were barely transported through the SLM to the strip solution. When the TBP concentration was increased to 40% and 60% (*v*/*v*), both the Nd and La recovery increased. After 6 h, the final Nd recovery was 24.98% and 34.15% for 40% and 60%, respectively, while the corresponding La recovery was similar at 25.20% and 34.95%. The decreasing recovery rate over time and the low final recovery of Nd and La, especially for low-concentration TBP, could be explained by the solubility of TBP in water and in the presence of nitrate ions. Firstly, the solubility of TBP in water is 400 mg/L, much higher than that of 204P (<100 mg/L) and 507P (<10 mg/L) [31]. Secondly, a significant amount of TBP could be bound to nitric acid, primarily from the strip solution, due to the formation of TBP-nitric acid complexes [32]. The equilibrium equations of TBP complexation of nitric acid can be written as
(8)$${\mathrm{HNO}}_{3}\left(\mathrm{aq}\right)+\mathrm{TBP}\;\left(\mathrm{aq}\right)\;↔\mathrm{TBP}\cdot {\mathrm{HNO}}_{3}\left(\mathrm{aq}\right)$$
(9)$${\mathrm{HNO}}_{3}\left(\mathrm{aq}\right)+\mathrm{TBP}\cdot {\mathrm{HNO}}_{3}\left(\mathrm{aq}\right)\;↔\mathrm{TBP}\cdot 2{\mathrm{HNO}}_{3}\left(\mathrm{aq}\right)$$

After combining with nitric acid, the ability of TBP to carry over REEs through SLM was significantly reduced. Finally, the solubility of TBP in nitric acid is likely much higher than the 400 mg/L in water due to the formation of TBP-nitric acid complexes, further depleting the SLM of TBP.

From Figure 3a,b, the Nd and La recovery rates both increase with increasing TBP concentration. Compared to Nd/La recovery by 507P and 204P, the selectivity of the two REEs using TBP was poor (Figure 3c). The nearly identical recovery of La and Nd using 40% and 60% (*v*/*v*) TBP resulted in selectivities of 1.00 and 0.98, respectively.

In summary, it is evident that the Nd recovery rate sequence was 507P > 204P > TBP. After 6 h, the Nd recovery was 88.80, 63.49, and 34.15, respectively. On the other hand, the La recovery was the opposite for the 3 extractants: TBP > 204P > 507P, 13.70, 20.82, and 34.95, respectively. Interestingly, 507P has the highest recovery rate for Nd and the lowest for La. These opposite recovery trends with the extractant resulted in 507P having the highest selectivity of 6.48, compared to 3.05 with 204P and 0.98 with TBP-HNO_3_.

### 3.3. The Effect of Different Types of Strip Solution

Cl^−^, SO_4_^2−^, and NO_3_^−^ are the most common anions for hydrometallurgical metal products, including leaching, separation, and other processes. HCl, H_2_SO_4_, and HNO_3_ have advantages and disadvantages in different recovery processes; therefore, it is crucial to optimize the acid system for each metal recovery and separation. For example, H_2_SO_4_ is most widely used for leaching on a large scale, but suffers from slow kinetics and precipitation; HCl has superior leaching efficiency, but corrosion increases capital cost and is expensive compared to H_2_SO_4_; HNO_3_ is a strong oxidant, but it decomposes and is also more expensive than H_2_SO_4_. Therefore, the use of an alternative anion in the strip phase is sometimes advantageous for mineral processing treatment. REE sulfates are the main REE product from bastnäsite and monazite leaching. However, it is necessary to transform to REE chlorides to achieve the best separation coefficient and processing efficiency. In this part of our work, different strip solutions with the same H^+^ concentration were used to explore the Nd/La transformations and their separation. Because the best selectivity was found with a 507P concentration of 5% (*v*/*v*) [26], the following tests were conducted with 5% (*v*/*v*) 507P feed solution with an initial pH of 3 in HCl.

Figure 4 shows the different acids in the strip had no noticeable effect on Nd transport. After 6 h, the Nd recovery was approximately 51% for all types of strip acid. The La recovery showed a dependence on the strip acid; recovery % was 1.17 for HCl strip solution, 1.87 for the H_2_SO_4_ strip solution, and 2.20 for the HNO_3_ strip solution. Based on the selectivities shown in Figure 4c, HCl had the best selectivity among the three strip solutions, at 44.25. More significantly, after the RE-Cl in the feed was transformed into RE-SO_4_ or RE-NO_3_ in the respective strip solutions, negligible Cl^-^ from the feed was detected in the strip phase. Liquid chromatography showed less than 2 mg/L of Cl^-^ in the strip solutions. The anion transformation and separation of Nd/La happen simultaneously, which could significantly simplify the REEs recovery and separation process.

### 3.4. The Effect of Auxiliary Agent

From previous research, the Nd and La transport rate increased significantly when the pH was greater than 3. However, throughout the process, the pH of the feed solution decreased monotonically, which may have impeded Nd and La transport. In this part, the role of an auxiliary agent, citric acid as a buffer, was investigated in the FSSLM Nd/La separation process. The feed phase was a pH 3 HCl solution, with 1000 mg/L La(III) and Nd(III) each, and 0, 0.2, and 0.5 M of citric acid. The strip phase was a 3 M HCl solution. The extractant was 20% (*v*/*v*) 507P. During the SLM tests, the pH was almost constant at 3 with 0.2 and 0.5 M citric acid. The Nd/La recovery and selectivity results are shown in Figure 5.

It was found that increasing the citric acid concentration from 0 to 0.5 M significantly reduced the Nd(III) recovery from about 88% at 0 M to 34.47% at 0.5 M. On the contrary, the addition of citric acid increased the recovery of La. The La recovery after 6 h was 13.70, 29.69, and 15.78% for 0, 0.2, and 0.5 M citric acid addition. As for the selectivity, it was found that citric acid reduced the selectivity for Nd recovery from Nd/La mix solution. The selectivity decreased from 5.90 with no citric acid to 2.24 and 2.18 with 0.2 and 0.5 M citric acid, respectively.

### 3.5. Stability of SLM

While SLM has been intensively studied mainly for separation technology, very few large-scale applications were found due to insufficient stability of the extractant in the membrane support. The loss of carrier and solvent from the membrane can generally be found from less than 1 h to a few months [33]. The degradation of SLM originates from: a. wetting of the pores in the membrane by the aqueous phases; b. the pressure difference between the membrane; c. emulsion formation or a carrier complex precipitation, which causes blockage of the membrane pores [34]. In this research, Nd/La separation was carried out for six days to evaluate SLM stability. The experimental conditions were: feed solution with Nd and La, 6 g/L each, initial pH of 3 HCl solution, strip solution was 3 M HCl, mixing rate was 350 rpm, and the extractant was 40% (*v*/*v*) 507P. Results are shown in Figure 6.

As can be seen from the graph, the Nd and La recovery still functioned with the FSSLM after six days. The Nd recovery was nearly stable after five days, with a recovery of 91%. On the other hand, the recovery of La was still increasing at six days, attaining 54%. Further examination of the SLM’s mass proved the carrier’s loss was negligible. The results confirm that the SLM has the potential for upscaled application for REEs recovery and separation.

## 4. Conclusions

This work demonstrated an effective and efficient Nd/La recovery and separation via flat sheet-supported liquid membrane in different extractant-acid systems, including 507P-HCl, 204P-H_2_SO_4_, and TBP-HNO_3_. Other parameters were tested as well to determine the optimal Nd separation from Nd/La mixtures, including the types of strip solution and an auxiliary agent. The results indicated that 507P-HCl was the most efficient extractant for both Nd recovery and Nd/La separation. The highest selectivity of 44.25 was obtained at the lowest 507P concentration of 5% (*v*/*v*) in D80. Concerning the different strip solutions, the greatest Nd/La was achieved with HCl as the strip solution, rather than H_2_SO_4_ or HNO_3_. The addition of citric acid in the feed solution kept the pH constant throughout the test, but reduced the Nd recovery from 88% at 0 M to 34.47% at 0.5 M after 6 h. A stability test proved that FSSLM still functioned well when extending the tests from 6 h to 6 days.

## Figures and Tables

**Figure 1 membranes-12-01197-f001:**
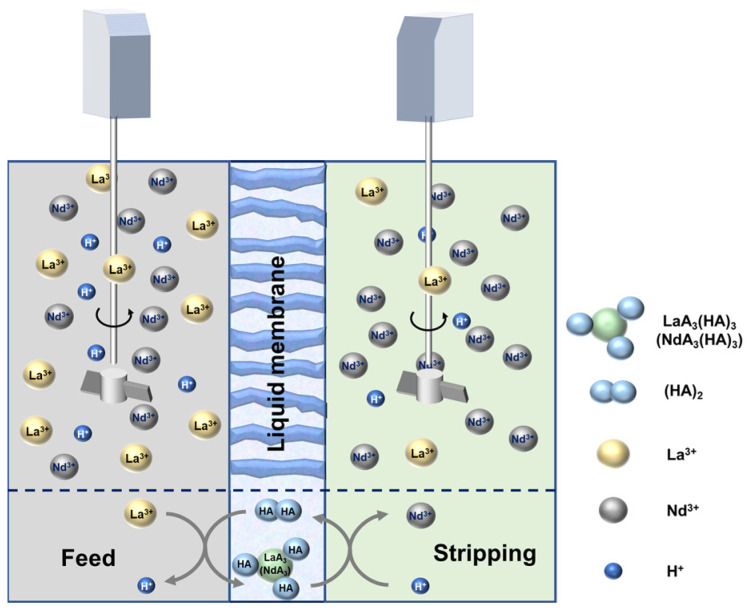
Schematic diagram of the FSSLM cell.

**Figure 2 membranes-12-01197-f002:**
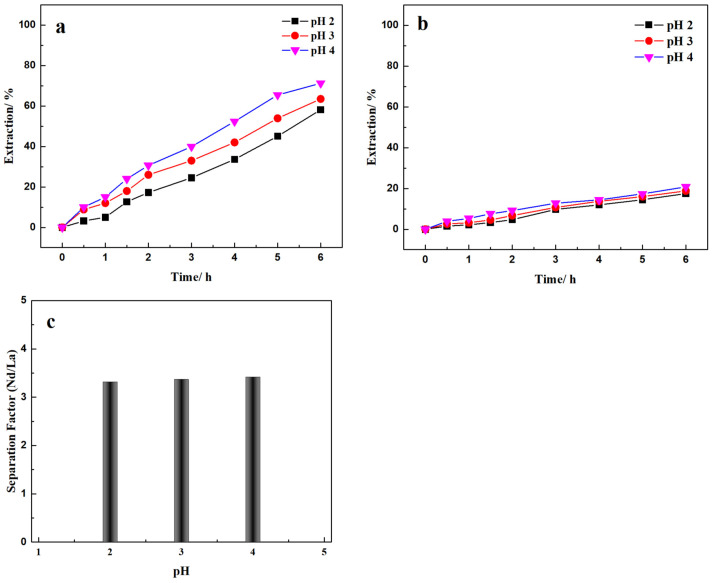
Effect of feed solution pH on (**a**) Nd and (**b**) La recovery and (**c**) separation. Feed: 1000 mg/L La(III) and Nd(III) in pH = 2, 3, 4 H_2_SO_4_ solutions. Extractant: 20% (*v*/*v*) 204P; Strip: 1.5 M H_2_SO_4_ solution.

**Figure 3 membranes-12-01197-f003:**
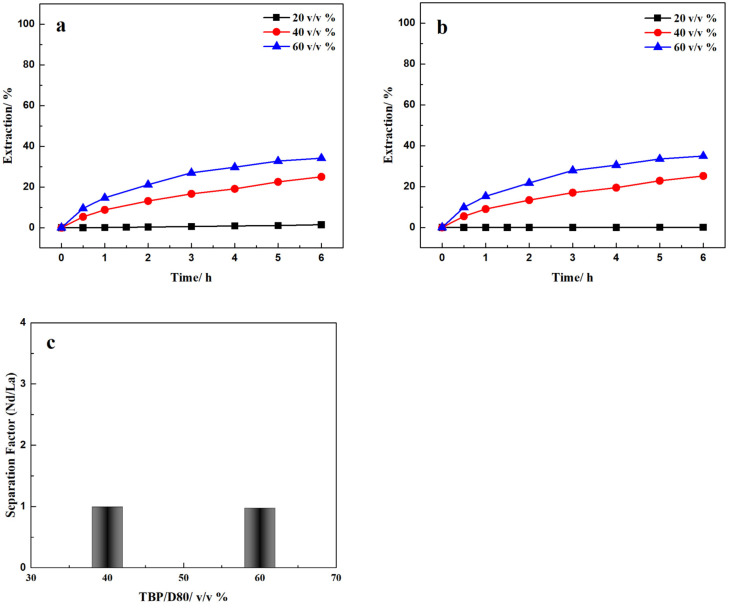
Effect of extractant on (**a**) Nd and (**b**) La recovery and (**c**) separation. Feed: 1000 mg/L La(III) and Nd(III) in pH = 3 HNO_3_ solutions Extractant: 20%, 40%, and 60% (*v*/*v*) TBP; Strip: 3 M HNO_3_ solution.

**Figure 4 membranes-12-01197-f004:**
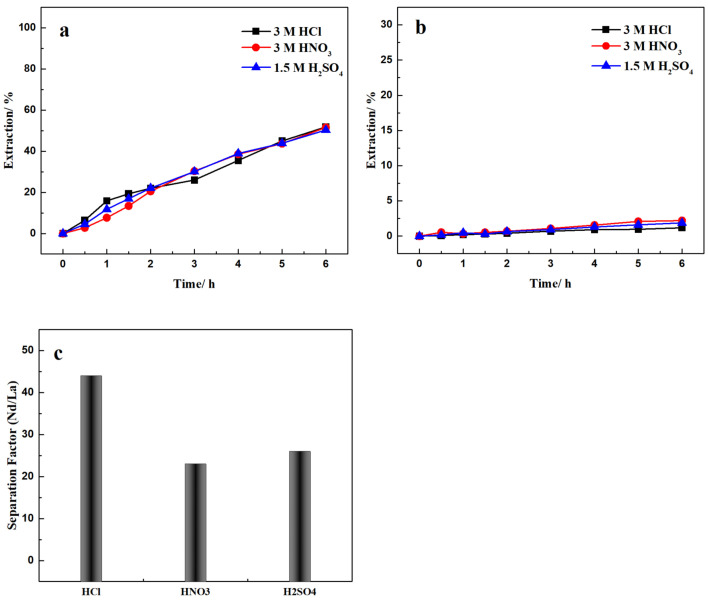
Effect of type of strip solution on (**a**) Nd and (**b**) La recovery and (**c**) separation. Feed: 1000 mg/L La(III) and Nd(III) in pH = 3 HCl solutions Extractant: 5% (*v*/*v*) 507P; Strip: 3 M HCl, HNO_3_ and 1.5 M H_2_SO_4_.

**Figure 5 membranes-12-01197-f005:**
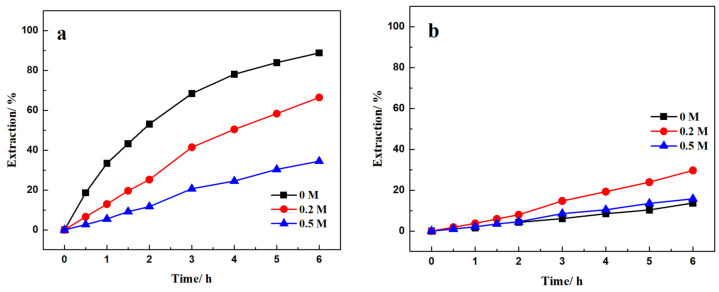
Effect of citric acid on (**a**) Nd and (**b**) La recovery and (**c**) separation. Feed: 1000 mg/L La(III) and Nd(III) in pH = 3 HCl solution with 0, 0.2 and 0.5 M citric acid Extractant: 20% (*v*/*v*) 507P; Strip: 3 M HCl solution.

**Figure 6 membranes-12-01197-f006:**
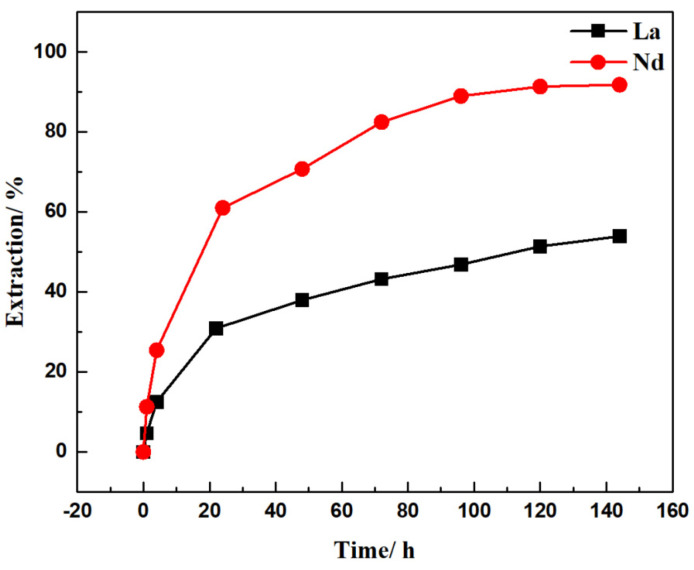
Nd and La recovery for extended to six days. Feed: 6000 mg/L La(III) and Nd(III) in pH = 3 HCl solution Extractant: 40% (*v*/*v*) 507P; Strip: 3 M HCl solution.

## Data Availability

The data presented in this study are available on request from the corresponding author.

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
