# Peer review of "Separation of Neodymium (III) and Lanthanum (III) via a Flat Sheet-Supported Liquid Membrane with Different Extractant-Acid Systems"

_membranes, 2022, doi:10.3390/membranes12121197_

Round 1
Reviewer 1 Report
In this work, the separation of neodymium (III) and lanthanum (III) in via a flat sheet supported liquid membrane was reported. It's important to develop separation method with high performance for the separation of different REEs. However, the scientific writing of this manuscript didn't reach the level of Membranes. I hope the following suggesting may help the authors to improve their manuscript.
1. As a membrane separation process, the flux of the solute should be provided.
2. Authors should pay more attention to the normative writing such as subscripts and subscripts.
3. It is recommended to provide a schematic diagram of the extraction process and device.
4. The extraction maintained a good linearity from 0 to 6 hours in Fig.1a,b. I'm curious if this linear increase will continue after 6 hours? Furthermore, why a linear relationship occurs should be discussed.
5. In Fig.2, the relationship of extraction over time is very different from Figure 1 and the reasons for this should be discussed carefully.

Author Response
Please see the attachement.

Reviewer 3 Report
Results obtained are well aligned to the objectives of the study. However, please address the following matters:
- Use of English proofreading service is REQUIRED. There are many grammatical errors, wrong sentence structure, and several sentences are vague and/or confusing.
- Title sentence structure is incorrect; please make correction
- Line 58-59: TBP is not an acidic organophosphorus extractant
- For manufacturer of chemicals, materials and equipment, information on city/state, country of manufacturer needs to be included.
- Line 87: the term “efficiencies” is more suitable instead of “rates”
- Line 97: The term “distribution ratio” could be confused with the term “distribution ratio” used in solvent extraction, please check.
- Eq. 5 – 6: The term extraction usually refers to extraction of metal from feed into solvent phase only; to indicate the efficiency of metal being transported into the stripping phase, the term “recovery” may be more suitable instead.
- Suggest rescaling separation factor axis on Figure 1(c) and 2(c)
- “Results and discussion” fall under section 3.
- Section 4. should be “Conclusion”, not “Discussion”
- Has extractant P507 been tested with H2SO4 feed? The experimental results do not seem to indicate the effect of acid in feed. The role of the extractant could be more significant. Stating that P507 was the most efficient extractant is acceptable but including the term “P507-HCl” could mislead readers on the importance of HCl in feed unless it is true. Please verify.
Round 2
Reviewer 1 Report
The Authors have addressed all of my concerns with the original manuscript. The revised manuscript is ready for publication.